**Data availability statement:** All relevant data are within the manuscript and its Supporting information files.

# Why do Chinese migrant workers return? Exploring economic push-pull factors and emotional ties

**Qiong Zhou[1], Zhe Huang[1], Lining Zeng** 🔵 **[2]\*, Jun Xu[3]\***

**1** College of Public Administration and Law, Hunan Agricultural University, Changsha, Hunan, China, **2** College of Business Administration, Hunan University of Finance and Economics, Changsha, Hunan, China, **3** College of Sports, Hunan University of Finance and Economics, Changsha, Hunan, China

\* zenglining@hufe.edu.cn (LZ); kofxujun@163.com (JX)

## Abstract

Over the past few decades, China's economic growth and urbanization have driven a significant migration of rural laborers to cities. Recently, however, an increasing number of migrant workers have chosen to return to their hometowns for employment opportunities. Understanding the factors influencing this return migration is crucial but challenging due to the complexity and diversity of these factors and their intricate interrelationships. Moreover, existing research on migrant workers' return lacks a systematic theoretical framework and comprehensive empirical analysis. To address these gaps, our study utilizes the "Push-Pull Theory" from migration theory to develop a comprehensive model. This model investigates how perceived benefits, trust, costs, and both personal and government support affect migrant workers' willingness to return. We employ structural equation modeling (SEM) for empirical analysis, revealing that perceived benefits, trust, and costs significantly influence migrant workers' perception of return support. This perception, in turn, enhances their willingness to return. Additionally, our findings show that government support positively moderates the relationship between perceived benefits and costs with return support. However, it does not significantly affect the relationship between perceived trust and support, indicating that policy incentives alone may not sufficiently build trust in hometowns. Furthermore, emotional factors—such as family and place attachment, community involvement, and quality of life in hometowns—indirectly influence the decision to return by shaping perceived benefits, trust, and costs. This study advances the application of Push-Pull Theory by integrating economic factors with emotional bonds in the context of return migration. It provides novel insights into how both economic incentives and emotional ties drive migrant workers' decisions to return, offering a more nuanced understanding of migration dynamics in China.

**Funding:** This study was financially supported by the Hunan Provincial Department of Education in the form of grants (XJJG-2024-039 and HNJG-2021-1143) received by QZ. No additional external funding was received for this study.

**Competing interests:** The authors have declared that no competing interests exist.

# 1 Introduction

In recent years, China has undergone rapid economic growth and urbanization, leading to a significant migration of rural laborers to urban areas [1]. By 2010, the migrant worker population had reached 242 million, with 122.64 million working outside their home regions, reflecting a 6% annual increase (National Bureau of Statistics, 2010) (source). However, recent trends indicate a shift in migration patterns, with more migrant workers opting for local or nearby employment. According to the 2020 Monitoring and Survey Report on Migrant Workers, the total number of migrant workers in 2020 was 285.6 million, a decline of 1.8% from the previous year. Among these, 169.59 million worked outside their home regions, marking a 2.7% decrease, while 116.01 million worked locally, a slight drop of 0.4% (source). This shift underscores a growing preference for jobs closer to home, driven by the need to balance family responsibilities and career opportunities. Several factors contribute to this trend. First, economic growth in China's central and western regions has accelerated, driven by industrial transfers and upgrades, which have created more local job opportunities. For instance, the GDP of China's western region now accounts for 21.5% of the national total, with industrial added value rising from 5.8 trillion yuan in 2019 to 8.1 trillion yuan in 2023. Concurrently, residents' per capita disposable income increased from 24,000 yuan in 2019 to 31,000 yuan in 2023, and the urbanization rate of permanent residents reached 59.94% (National Bureau of Statistics, 2023) (source). Second, improvements in rural infrastructure and public services have enhanced living conditions, making local employment more attractive. Finally, family considerations, such as childcare and education, play a pivotal role in motivating migrant workers to seek jobs closer to home.

The Push-Pull Theory, initially proposed by E.S. Lee in the 20th century, provides a foundational framework for understanding migration dynamics [2]. Lee categorized migration influences into "push" factors, such as poverty and unemployment, which drive individuals away from their place of origin, and "pull" factors, such as better job opportunities and improved living conditions, which attract individuals to new destinations. While this theory has been widely applied to various migration contexts, including rural-to-urban and international migration, its application to the study of migrant workers' return intentions remains limited. Existing research, such as Bi et al. (2019), has focused on social networking among migrant workers [3], while Fan et al. (2025) has examined land ownership and labor economy dynamics [4]. However, these studies have not fully explored the interplay between economic incentives and emotional ties in shaping return migration decisions. This study addresses this gap by integrating economic push-pull factors with emotional ties, such as family attachment, community involvement, and quality of life, to provide a more comprehensive understanding of return migration [5]. Unlike previous studies, which have primarily focused on urban adaptation and settlement, our research emphasizes the role of emotional bonds in influencing migrant workers' decisions to return. By constructing a model based on the Push-Pull Theory, this study aims to identify key factors influencing return intentions and examine their interactions. Additionally, this study includes government support as a moderating variable to assess its impact on these relationships. Government policies, such as subsidies for returning entrepreneurs, vocational training programs, and rural infrastructure investments, can amplify the economic appeal of returning home while also addressing emotional factors, such as improved quality of life and family reunification. Through a quantitative survey, this study gathers empirical data to test our model, offering valuable insights for policymakers seeking to promote rural economic growth and revitalization by encouraging migrant workers to return [6].

## 2 Theoretical background

### 2.1 The concept of migrant worker migration

Population migration is the movement of individuals or groups across geographic regions, driven by diverse economic, social, political, and environmental factors [7]. Migrant worker migration, a significant subset of population migration, specifically refers to the movement of rural laborers from rural areas to urban regions [8]. In urban areas, migrant workers are primarily involved in non-agricultural production and service activities. This migration profoundly impacts labor distribution between rural and urban areas, economic development, and social structures [9]. Migrant worker migration not only reshapes labor market supply and demand but also accelerates urbanization. Additionally, it fosters rural economic transformation and upgrading [10].

This study focuses on the return intentions of migrant workers, referring to the tendency of those employed outside their hometowns to return for work and residence. Such return behavior reflects migrant workers' overall evaluation of their hometowns versus their current place of residence. Understanding this behavior is essential for analyzing current migration dynamics and rural development trends. An in-depth analysis of return intentions offers insights into the underlying mechanisms of population mobility. It also provides a theoretical foundation for developing effective policy measures.

### 2.2 The "Push-Pull Theory" in migration theory

The Push-Pull Theory, introduced by Everett S. Lee in the 1960s, serves as a foundational framework in migration studies [11]. This theory categorizes migration drivers into "push" and "pull" factors. Push factors refer to adverse conditions in the place of origin, such as poverty, unemployment, resource scarcity, and environmental degradation, which compel individuals to migrate. Conversely, pull factors represent favorable conditions in the destination, including better job opportunities, improved quality of life, and enhanced social welfare, which attract migrants [12]. Six key factors that drive their location settlement decisions have been identified regarding Chinese settlement preferences. These include educational resources, public transportation, infrastructure and amenities, economic and employment opportunities, environment, and safety [13–15]. Additionally, Lee emphasized the role of intervening obstacles—such as distance, cost, policy restrictions, and information asymmetry—as well as individual characteristics like age, education level, family status, and personality traits, in shaping migration decisions [16]. Over the decades, the Push-Pull Theory has been widely applied to various migration contexts, including rural-to-urban migration, international migration, and disaster-induced displacement [17]. For instance, studies by Kwilinski et al. (2022) and Sanliturk et al. (2023) have utilized the theory to explain labor migration patterns in developing countries, highlighting the interplay between economic disparities and migration flows [18,19]. According to Hormozinejad et al.'s (2024) research, Chinese immigrants in Sydney consider tangible factors such as educational resources, public transportation, infrastructure, economy, environment, and safety when choosing suburbs, as well as intangible factors such as cultural identity and Chinese cultural atmosphere. The Chinese would choose to settle in an area because of the advantageous characteristics exhibited in the suburb [20].

Despite its broad applicability, the Push-Pull Theory has been underexplored in the context of migrant workers' return intentions. Existing research has predominantly focused on the motivations and adaptation processes of migrant workers in urban areas, with limited attention to the factors driving their decisions to return to their hometowns [21]. This gap

is particularly notable given the recent emergence of return migration as a significant phenomenon in China. As the country undergoes rapid economic development and urbanization, the complexity of migrant workers' return intentions has become increasingly apparent. These intentions are influenced by a combination of economic, social, and emotional factors, including economic benefits, family responsibilities, cultural identity, and social support [22]. For instance, rising urban living costs, heightened job competition, and challenges in social integration have emerged as new push factors, discouraging migrant workers from remaining in cities. On the other hand, opportunities for local development, strong family ties, access to land, and cultural connections in their hometowns act as pull factors, encouraging their return [23].

Recent advancements in technology and social networks have further complicated these dynamics. The proliferation of digital platforms and mobile communication tools has enabled migrant workers to maintain stronger connections with their hometowns, thereby influencing their return decisions [24]. Social networks, both online and offline, play a dual role: they provide critical information about job opportunities and living conditions in both urban and rural areas, while also fostering emotional bonds that can either reinforce or mitigate the push-pull factors [25]. For example, social media platforms allow migrant workers to stay informed about local economic developments and community events, which can enhance the pull of their hometowns. Simultaneously, these platforms facilitate emotional support networks that may alleviate the sense of isolation in urban areas, potentially reducing the push to return [26].

This study extends the Push-Pull Theory by integrating emotional bonds—such as family attachment, community involvement, and a sense of belonging—into the framework. While traditional applications of the theory have focused on economic and environmental factors, the inclusion of emotional ties enhances its explanatory power in the context of return migration. Emotional bonds not only act as pull factors but also moderate the influence of economic incentives on migration decisions. For example, strong family ties may amplify the pull of local job opportunities, while a lack of social integration in urban areas may reinforce the push to return home. By incorporating these dimensions, our study provides a more nuanced understanding of the interplay between economic and emotional factors in shaping migrant workers' return intentions. Furthermore, this refined framework is particularly relevant to China's unique socio-economic context, where rapid urbanization and rural revitalization policies have created dynamic push-pull dynamics. By applying this extended framework, our study not only contributes to the theoretical advancement of migration studies but also offers practical insights for policymakers aiming to promote urban-rural coordination and sustainable rural developmen.

## 3 Hypotheses

### 3.1 Conceptual model

This study employed the Likert five-point scale to quantify the relationships among variables. Specifically, family attachment assessed the emotional bond between individuals and their hometown relatives; local emotions gauged identification and attachment to hometown culture, environment, and traditions. Community involvement was measured by participation frequency, role engagement, and social network extent. Quality of life in hometown encompassed dimensions such as healthcare, education, leisure, and public security. Perceived benefits involved expectations regarding economics, quality of life, work-family balance, and personal development. Perceived trust pertained to trust in the government, policies, and social environment, while perceived costs encompassed economic, time, and psychological burdens.

Support was quantified by attitudes towards rural tourism, construction, environment, and policies. Government support was evaluated through satisfaction with return policies, subsidies, training, and infrastructure improvements. The intention to return home reflected individuals' intentions to work, live, and settle in their hometown. The conceptual model is shown as Fig 1:

## 3.2 Relationship between support and intention to return home

In recent years, the support of migrant workers for the tourism industry has received widespread academic attention. Support level refers to the attitude of migrant workers towards returning home to develop their careers [27]. According to the theory of "cognition emotion intention" relationship, an individual's emotions are influenced not only by their cognition of things, but also by their behavioral intentions and actual behavior. Scholars have found through further research that perceived benefits play a significant role in the relationship between local emotions and community participation on support. According to social

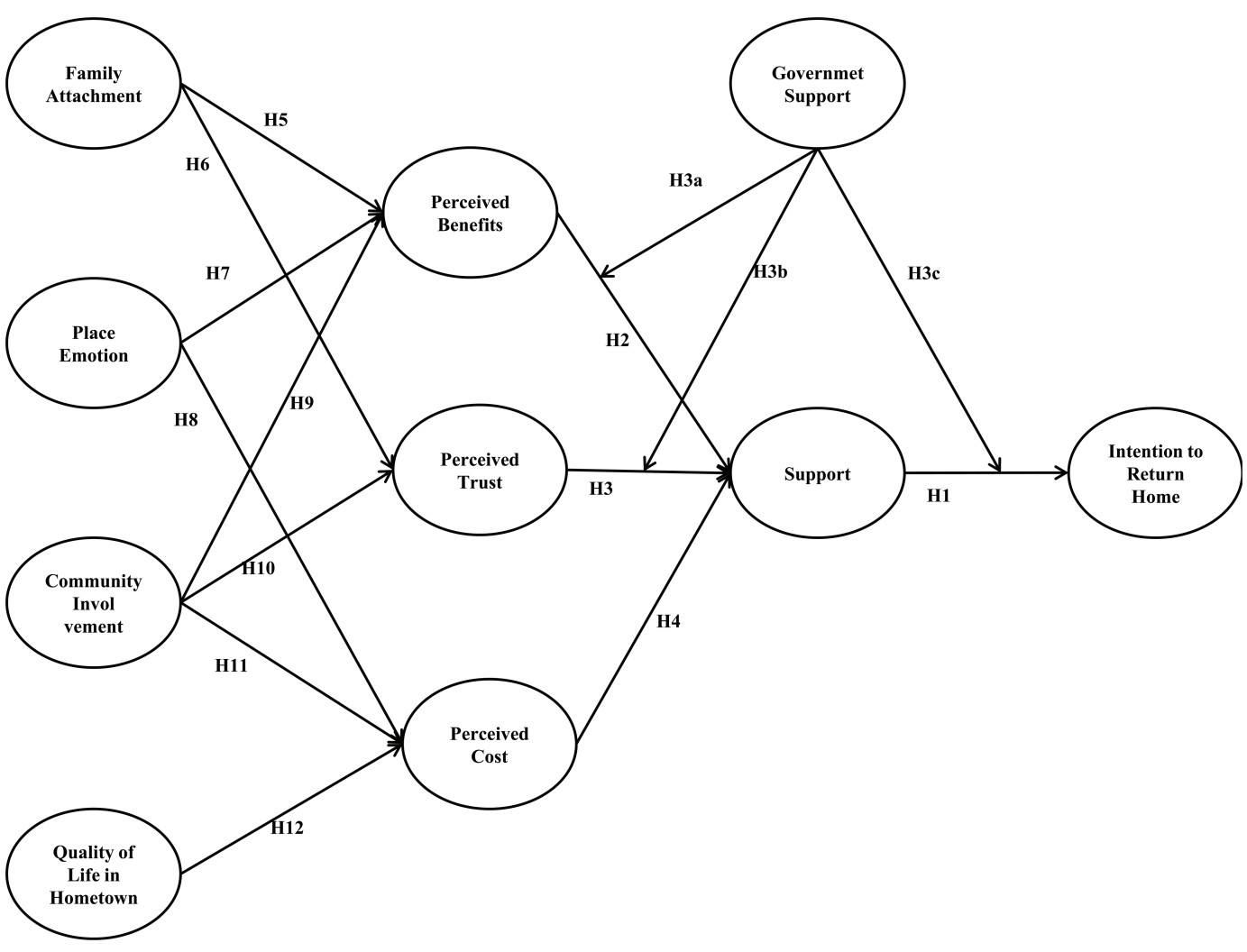

**Fig 1. Research model.**

exchange theory, if migrant workers perceive that the benefits of returning home outweigh the costs, they will hold a positive attitude towards returning home [28]. Support, as a pre psychological variable, affects the decision to return home through a dual pathway: on the one hand, high support directly strengthens the willingness to return home; On the other hand, it will also encourage migrant workers to actively seek specific information and opportunities for their return to their hometowns, thereby indirectly increasing the likelihood of actual return. Recent studies have also found that this relationship is moderated by individual characteristics (such as age, skill level) and situational factors (such as local policy intensity), indicating that the process of converting support to willingness to return home is conditionally dependent [29]. Hence, the following hypothesis is constructed:

H1: Support is positively correlated with the Intention to Return Home.

## 3.3 Relationship between perceived benefits and support

Perceived benefits are crucial in shaping support factors. For migrant workers, the perceived benefits of returning home primarily include economic advantages, improved quality of life, and emotional satisfaction. Economic benefits are a key consideration for migrant workers contemplating a return home. When migrant workers perceive increased job opportunities and income in their hometowns, they are more inclined to return. They view employment in their hometown as economically secure, enhancing their sense of government and social support [30]. The improvement of rural infrastructure and living conditions has enhanced the quality of life for migrant workers upon their return, increasing their perception of support for returning home [31]. Emotional satisfaction is another important component of perceived benefits. Reuniting with family and reintegrating into familiar communities provides important emotional support for migrant workers, further enhancing their desire to return home [32]. Therefore, the higher the economic benefits, quality of life, and emotional satisfaction that migrant workers feel, the stronger their willingness to return home for development. Hence, the hypothesis is formulated as follows:

H2: Perceived Benefits are positively correlated with Support.

## 3.4 Relationship between perceived trust and support

Perceived trust is essential in shaping support and primarily reflects migrant workers' trust in family, society, and government. Trust in family members enhances migrant workers' perception of family support. Trust strengthens emotional bonds, making migrant workers more reliant on family support [33]. Additionally, social trust is essential for social support after returning home. When migrant workers trust their hometown communities and neighborhood relations, they are more likely to integrate into society and gain community support [34]. Finally, trust in government is a core factor shaping migrant workers' perception of government support. Migrant workers are more likely to recognize government support and feel confident about returning when they have strong trust in government policies and public services [35]. Thus, the greater the trust migrant workers have in family, society, and government, the stronger their perceived support. Hence, the following hypothesis is constructed:

H3: Perceived Trust is positively correlated with Support.

## 3.5 Relationship between perceived cost and support

Perceived cost is a significant factor influencing migrant workers' perception of support. It primarily encompasses economic, time, and psychological costs. Economic cost is a primary

concern for migrant workers considering a return home. Transportation expenses, relocation costs, and potential income reductions associated with returning influence their perception of support [36]. When the perceived cost of returning home is low, migrant workers are more likely to find local job opportunities and government support appealing, thereby increasing their recognition of support [37]. Time cost is another essential component of perceived cost. Shorter time required for returning home enables migrant workers to more easily gain family and community support, reducing concerns about returning [38]. Psychological cost also significantly influences the perception of support. Returning home may require letting go of an urban lifestyle and readapting to rural life. If migrant workers perceive improvements in their hometown environment and feel emotional support from family and society, their psychological burden decreases, enhancing their perception of support [39]. Hence, the hypothesis formulated as follows:

H4: Perceived Cost is positively correlated with Support.

## 3.6 Government support as a moderator of other relationships

In the process of migrant workers returning home, government support acts as both a direct influencing factor and a significant moderating variable between other factors and support. Government support, through policies, funding, and services, can significantly enhance migrant workers' perception of benefits in their hometowns. Policies related to entrepreneurship, land use, training, and employment subsidies increase migrant workers' perception of economic benefits, strengthening the positive relationship between perceived benefits and support [40]. High levels of government support make migrant workers more confident in achieving stable economic returns at home, enhancing the influence of perceived benefits on their perception of social support. Additionally, government support significantly enhances migrant workers' trust in both the government and society, moderating the relationship between perceived trust and support. When migrant workers see active government efforts in policy implementation, public service quality, and fairness, their trust in the government increases. This heightened trust strengthens their perception of support during the return process [41]. Government-provided subsidies, healthcare, and educational support reinforce migrant workers' belief that they will have adequate life support upon returning, strengthening their perception of support. Finally, government support plays a crucial role in reducing economic, time, and psychological costs for migrant workers returning home. For example, government-provided transportation subsidies, relocation assistance, and entrepreneurship support can effectively reduce the economic burden of returning, enhancing migrant workers' perception of family and social support [42]. Additionally, government-provided information and policy consultation reduce uncertainty and information asymmetry, lowering psychological costs. As a key moderating variable, government support enhances perceived benefits, increases trust, and lowers costs, thus influencing the relationship between these factors and support. This serves as an important foundation for migrant workers' perception of support during their return process. Hence, the following hypotheses are constructed:

H3a: Government Support moderates the relationship between Perceived Benefits and Support.

H3b: Government Support moderates the relationship between Perceived Trust and Support.

H3c: Government Support moderates the relationship between Perceived Cost and Support.

## 3.7 Relationship between perceived benefits and other factors

The perceived benefits of returning home for migrant workers are influenced by factors such as family attachment, place attachment, and community involvement. Family attachment significantly affects migrant workers' perception of benefits. Emotional bonds with family provide migrant workers with emotional satisfaction and support, enhancing their perception of potential economic and life benefits [43]. When migrant workers have strong family attachment, they place higher value on the emotional benefits of family reunion, directly enhancing their perception of the benefits of returning. Place attachment reflects migrant workers' sense of belonging and emotional connection to their hometown. As hometown infrastructure and living conditions improve, migrant workers' sense of identity and belonging to their hometown strengthens, increasing their perception of the benefits of returning [44]. This attachment fosters hope for a better quality of life and economic opportunities upon returning. Community involvement reflects the degree to which migrant workers integrate and participate in community activities after returning. Engaging in community-building activities strengthens migrant workers' connections and emotional bonds within the community, fostering greater social support and life satisfaction, which enhances their perception of the benefits of returning [45]. Community involvement helps migrant workers adjust to life after returning and provides a sense of belonging and social recognition, important sources of perceived benefits. Family attachment, place attachment, and community involvement are crucial factors shaping migrant workers' perception of benefits. By strengthening emotional bonds and social connections, these factors increase migrant workers' awareness and expectations of the benefits of returning. Hence, the hypotheses formulated as follows:

H5: Family Attachment is positively correlated with Perceived Benefits.

H7: Place Attachment is positively correlated with Perceived Benefits.

H9: Community Involvement is positively correlated with Perceived Benefits.

## 3.8 Relationship between perceived trust and other factors

Family attachment and community involvement are key factors influencing migrant workers' perceived trust, as they strengthen emotional connections with family and community. Family attachment represents a deep emotional bond migrant workers feel toward family members. Family support and emotional bonds significantly increase migrant workers' trust in their families [46]. Migrant workers with strong family attachment are more likely to trust family members and view them as a reliable source of support [47]. This attachment boosts trust in family members, further strengthening migrant workers' confidence in returning home and their perception of support. Community involvement is also a key factor influencing migrant workers' perceived trust. Active participation in community activities, integration into the community, and building good relationships with neighbors gradually increase migrant workers' trust in the community [48]. Community involvement allows migrant workers to experience care and support from community members, strengthening trust in the community and providing emotional satisfaction and a sense of security. Hence, the following hypotheses are proposed:

H6: Family Attachment is positively correlated with Perceived Trust.

H10: Community Involvement is positively correlated with Perceived Trust.

## 3.9 Relationship between perceived cost and other factors

Place attachment, community involvement, and hometown quality of life are key factors influencing migrant workers' perceived costs. These factors impact perceptions of costs

encountered during the return process by enhancing emotional connections, community integration, and quality of life. Place attachment refers to migrant workers' emotional connection and sense of belonging to their hometown. Strong emotional attachment to their hometown increases migrant workers' willingness to bear costs associated with returning [49]. This attachment makes them more willing to accept economic costs like transportation and relocation [50]. Community involvement is another key factor affecting perceived costs. Active participation in community activities provides migrant workers with emotional support and social recognition from community members, reducing concerns about the costs of returning [51]. Greater community involvement decreases perceived psychological and time costs associated with returning and helps migrant workers approach the challenges of returning more positively. Hometown quality of life is another important factor influencing perceived costs. Improved hometown infrastructure and public services significantly enhance quality of life, reducing perceived economic and psychological costs of returning [52]. A high-quality living environment provides better conditions for returning migrant workers, alleviating their sense of burden related to return costs. Hence, the hypotheses formulated as follows:

H8: Place Attachment is positively correlated with Perceived Cost.

H11: Community Involvement is positively correlated with Perceived Cost.

H12: Quality of Life in Hometown is positively correlated with Perceived Cost.

## 4 Experimental results

### 4.1 Source of questionnaire items

The questionnaire in this study was designed to explore key factors influencing migrant workers' intentions to return home, with a focus on perceived benefits, trust, support, and return costs. Grounded in the Push-Pull Theory from migration studies and informed by empirical research, the questionnaire ensures scientific rigor and validity. Table 1 presents all measurement items and their sources. The questionnaire assesses push factors, including perceived costs and trust, which measure negative conditions that prompt migrant workers to leave cities and consider returning home. It also evaluates pull factors, such as family attachment, place attachment, community involvement, quality of life, perceived benefits, government support, and return intention, to gauge the extent to which migrant workers perceive economic opportunities, quality of life, and social support in their hometowns. The questionnaire incorporates items from classic theoretical models and recent research on migrant worker mobility and policy contexts.

This study adopted a mixed mode sampling strategy that combines online and offline methods to ensure comprehensive coverage of the views of migrant workers in Hunan Province. For the offline part, a stratified purposive sampling method is used to distribute paper questionnaires in public places (including train stations, labor markets, urban-rural transition zones, and manufacturing enterprises) to capture the diversity of key demographic data such as age, gender, industry sectors, and immigration duration. The online section utilized social media platforms such as WeChat, QQ, and the Migrant Workers Forum, and implemented snowball sampling to expand coverage through initial contact with non-governmental organization partners and trade unions.

From March to July 2024, 400 questionnaires (250 offline and 150 online) were distributed in 12 cities and counties in Hunan Province, with a total of 312 valid responses and a response rate of 78%. Although random sampling is logically challenging, some measures have been implemented to address potential biases: comparisons of early and late stage respondents showed no significant differences in key variables, and online responses were weighted to compensate for their lower response rate (70%) compared to offline responses (82%). The

Table 1. Sources of questionnaire measurement items.

| Push-Pull Theory Factor Classification | Constructs | Number of Measurement Items | Sources |
|---|---|---|---|
| **Push Factors : Negative factors encouraging migration** | Quality of Life in Hometown | 4 | Qing et al., 2022 [50]; Wu et al., 2023 [51] |
| | Perceived Cost | 5 | Conigliani et al., 2021 [12]; Belau et al., 2021 [38] |
| **Pull Factors : Positive factors attracting return** | Family Attachment | 4 | Ngan et al., 2023 [29];Chang et al., 2021 [49] |
| | Place Emotion | 5 | Lin et al., 2020 [34];Chang et al., 2021 [49] |
| | Community Involvement | 5 | Huyen et al., 2020 [30];Adedeji et al., 2023 [52] |
| | Perceived Benefits | 6 | Huyen et al., 2020 [30]; Kim et al., 2022 [32] |
| | Perceived Trust | 4 | Kim et al., 2022 [32]; Roslavtseva, 2023 [35] |
| | Support | 4 | Zhou et al., 2021 [23] ;Niu et al., 2023 [31] |
| **Moderator Factors : Variables influencing migration process** | Government Support | 5 | Alshaaban et al., 2021 [40]; Vaculovschi et al., 2023 [42] |
| **Outcome Factors : Direct results of migration decision** | Intention to Return Home | 4 | Duan et al., 2020 [5]; Hu et al., 2022 [27] |

method of stratification by industry and hometown helps to improve representativeness. In addition, we conducted post hoc checks to assess non response bias and compared the differences in key variables between early and late stage respondents, and found no significant differences.

Due to Hunan Province being the main source of migrant workers in China, it was selected as the research site. The significant outflow of labor force from the province to economically developed areas, especially the Pearl River Delta region, makes it a representative case for understanding the trend of returning immigrants. Its socio-economic characteristics, including a mix of rural and urban areas, are consistent with the socio-economic characteristics of many central and western provinces, which are also the main source of migrant workers. This makes Hunan an ideal case study for examining the factors influencing the return of investment in regions with similar economic and cultural backgrounds.

## 4.2 Analysis of basic information from the questionnaire

The questionnaire's basic information provides an analysis of the demographic and occupational distribution of participating migrant workers, covering gender, age, workplace, job type, family status, and years of work. Among the 312 respondents, Fig 2 shows that 59% were male (184) and 41% female (128), suggesting a larger representation of men within the migrant worker population. The largest age group was 36-45, with 127 respondents (40.7%), followed by those aged 26-35, comprising 27.9% (87 respondents). This distribution indicates that migrant workers are mostly middle-aged or young adults, especially individuals balancing family and career. Additionally, 42 respondents were aged 18-25, 39 were 46-55, and 17 were 56 or older, representing smaller age groups. In terms of workplace location, Shenzhen had the largest number of respondents (86), followed by Guangzhou (72), Chengdu (69), Hangzhou (62), and Shanghai (16). The smallest group was in Beijing, with only 7 respondents. The distribution suggests that migrant workers are primarily concentrated in the Pearl River and Yangtze River Delta regions, especially in cities like Shenzhen and Guangzhou. Guangzhou and Shenzhen, located in the Pearl River Delta, are geographically closer to Hunan and have historically attracted many migrant workers from the province

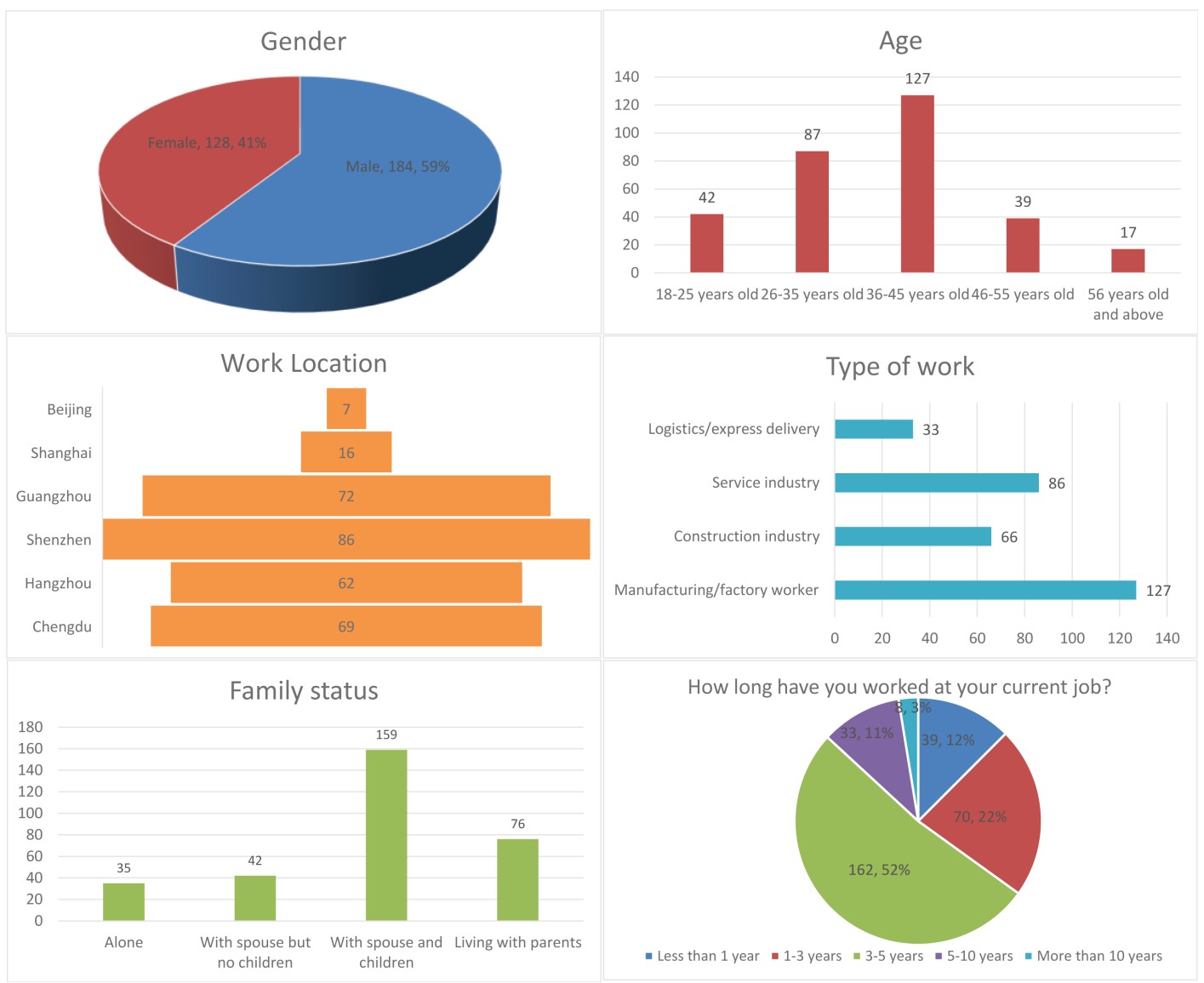

**Fig 2. Distribution results of basic information in the questionnaire.**

due to their booming manufacturing and service sectors. In contrast, Beijing, and Shanghai, while economically dominant, are farther away and have higher living costs, which may deter some migrant workers. According to the 2023 China Statistical Yearbook, the average monthly rent per square meter in Beijing (¥86.5) and Shanghai (¥82.1) was approximately 30% higher than in Guangzhou (¥62.3) and Shenzhen (¥65.8). Additionally, the ECA International 2022 Cost of Living Survey ranked Beijing and Shanghai among the top 10 most expensive Chinese cities, while Guangzhou and Shenzhen fell outside the top 15 [53].This pattern aligns with broader migration trends in China, where proximity and economic opportunities play a significant role in destination choices [54]. Among job types, the largest group was manufacturing/factory workers, comprising 127 respondents (40.7%), followed by the service industry (86 respondents, 27.6%), construction (66 respondents, 21.2%), and logistics/courier

(33 respondents, 10.6%). These results indicate that a high proportion of migrant workers are employed in manufacturing and service sectors. For family status, 159 respondents (51%) lived with their spouse and children, the largest group. Additionally, 76 respondents (24.4%) lived with parents, 35 (11.2%) lived alone, and 42 (13.5%) lived with a spouse but no children. These findings indicate that most migrant workers choose to live with family members in their place of work. Regarding years of work, the largest group had been in their current position for 3-5 years, with 162 respondents (52%). This was followed by those working 1-3 years (70 respondents, 22%), less than 1 year (39 respondents, 12%), and over 5 years (41 respondents, 13%). This concentration in the 3-5 year range suggests relatively high job stability among migrant workers.

## 4.3 Exploratory factor analysis

This study initially conducted a reliability analysis on each variable to assess the consistency and stability of the measurement items. Cronbach's alpha was calculated for each variable, and Table 2 shows that all variables have an alpha coefficient above 0.85, indicating high internal consistency. The overall alpha coefficient is 0.933, indicating high reliability for the entire questionnaire. To assess validity, the Kaiser-Meyer-Olkin (KMO) test was used to evaluate data suitability. The results show that all variables have KMO values above 0.80, indicating suitability for factor analysis. Place emotion and government support have the highest KMO values, both at 0.883. Perceived benefits have a KMO value of 0.906, indicating high data quality for this variable, which effectively explains the factor structure. These results indicate that the questionnaire has high reliability and validity.

Principal component analysis was used to extract the factor structure for each variable. Table 3 shows that ten factors with initial eigenvalues greater than 1 were extracted based on the total variance explained. The cumulative variance explained rate is 70.701%, indicating that these factors effectively account for the total variance in the variables. After rotation, the first four factors show high variance explained rates of 8.490%, 7.798%, 7.780%, and 7.775%, respectively. The cumulative variance explained rate for these four factors is 31.843%, highlighting their significance in the structure. The variance explained rates for the remaining factors gradually decrease. The fifth to tenth factors explain variances of 7.427%, 6.797%, 6.375%, 6.104%, 6.096%, and 6.058%, respectively. Although these factors contribute less to the overall structure, they still provide explanatory power. The ten extracted factors collectively explain most of the variance in the variables, confirming the scale's multidimensional structure and supporting the factor analysis results.

**Table 2. Reliability and effectiveness.**

| Variables | Items | Alpha | KMO |
|---|---|---|---|
| Family Attachment | 4 | 0.894 | 0.841 |
| Place Emotion | 5 | 0.890 | 0.883 |
| Community Involvement | 5 | 0.887 | 0.882 |
| Quality of Life in Hometown | 4 | 0.850 | 0.823 |
| Perceived Benefits | 6 | 0.885 | 0.906 |
| Perceived Trust | 4 | 0.863 | 0.825 |
| Perceived Cost | 5 | 0.878 | 0.869 |
| Support | 4 | 0.866 | 0.823 |
| Government Support | 5 | 0.895 | 0.883 |
| Intention to Return Home | 4 | 0.862 | 0.817 |
| **Total** | 46 | 0.933 | 0.902 |

**Table 3. Factor analysis results: total variance explained.**

| Item | Initial Eigenvalues | | | Rotated Loadings Sum of Squares | | |
|---|---|---|---|---|---|---|
| | Eigenvalue | Variance (%) | Cumulative (%) | Sum of Squares | Variance (%) | Cumulative (%) |
| 1 | 11.981 | 26.046 | 26.046 | 3.905 | 8.490 | 8.490 |
| 2 | 3.442 | 7.482 | 33.527 | 3.587 | 7.798 | 16.288 |
| 3 | 2.964 | 6.443 | 39.970 | 3.579 | 7.780 | 24.069 |
| 4 | 2.910 | 6.326 | 46.297 | 3.576 | 7.775 | 31.843 |
| 5 | 2.490 | 5.414 | 51.710 | 3.417 | 7.427 | 39.271 |
| 6 | 2.000 | 4.347 | 56.057 | 3.127 | 6.797 | 46.068 |
| 7 | 1.955 | 4.251 | 60.308 | 2.933 | 6.375 | 52.443 |
| 8 | 1.922 | 4.177 | 64.485 | 2.808 | 6.104 | 58.547 |
| 9 | 1.590 | 3.457 | 67.943 | 2.804 | 6.096 | 64.643 |
| 10 | 1.269 | 2.759 | 70.701 | 2.787 | 6.058 | 70.701 |

## 4.4 Confirmatory factor analysis

**4.4.1 Model fit index analysis.** Confirmatory factor analysis (CFA) was conducted to assess the model fit, as shown in Table 4. The CMIN/DF value is 1.252, below the threshold of 3, indicating a good model fit. Fit indices include Normed Fit Index (NFI), Relative Fit Index (RFI), Incremental Fit Index (IFI), Tucker-Lewis Index (TLI), and Comparative Fit Index (CFI), with values of 0.86, 0.85, 0.97, 0.96, and 0.97, respectively. IFI, TLI, and CFI values are all above 0.90, demonstrating an ideal model fit. NFI and RFI values are slightly below the 0.90 threshold, suggesting some room for improvement but not affecting the overall model fit. The Goodness of Fit Index (GFI) is 0.86, slightly below the 0.90 standard but close to the threshold, indicating an overall good model fit. The Root Mean Square Error of Approximation (RMSEA) is 0.028, below the 0.08 threshold, indicating a good fit. The CFA results indicate a good overall model fit, with most indices meeting the evaluation criteria.

**4.4.2 Convergent validity and discriminant validity analysis.** This study assessed convergent validity using Average Variance Extracted (AVE) and Composite Reliability (CR). As shown in Table 5, AVE values for each construct range from 0.562 to 0.678, all above the 0.5 threshold, indicating good convergent validity. This suggests that each construct effectively explains the variance of the latent variables. For CR values for each construct range from 0.850 to 0.894, all exceeding the 0.7 threshold. This indicates high internal consistency, showing that the measurement items effectively reflect the latent variables. Additionally, Table 6 shows that the correlation coefficients between constructs are less than the square root of their respective AVE, confirming discriminant validity. Thus, both convergent and discriminant validity are confirmed in this study.

## 4.5 Path analysis of the structural equation model

In the path analysis of the structural equation model, this study tested the hypothesized relationships, with results shown in Table 7. The results indicate that family attachment has no significant effect on perceived benefits (H5 not supported, P=0.116). However, family

**Table 4. Model fit of confirmatory factor analysis.**

| Model Fit | CMIN | DF | CMIN/DF | NFI | RFI | IFI | TLI | CFI | GFI | RMSEA |
|---|---|---|---|---|---|---|---|---|---|---|
| Fit Results | 1182.281 | 944 | 1.252 | 0.86 | 0.85 | 0.97 | 0.96 | 0.97 | 0.86 | 0.028 |
| Judgment Std. | - | - | <3 | >0.9 | >0.9 | >0.9 | >0.9 | >0.9 | >0.9 | <0.08 |

**Table 5. Convergent validity and composite reliability.**

| Construct | Item | Loading Factor | CR | AVE |
|---|---|---|---|---|
| Family Attachment | FA1 | 0.833 | 0.894 | 0.678 |
| | FA2 | 0.833 | | |
| | FA3 | 0.787 | | |
| | FA4 | 0.839 | | |
| Place Emotion | PE1 | 0.810 | 0.889 | 0.618 |
| | PE2 | 0.771 | | |
| | PE3 | 0.772 | | |
| | PE4 | 0.791 | | |
| | PE5 | 0.785 | | |
| Community Involvement | CI1 | 0.800 | 0.887 | 0.610 |
| | CI2 | 0.758 | | |
| | CI3 | 0.797 | | |
| | CI4 | 0.793 | | |
| | CI5 | 0.756 | | |
| Quality of Life in Hometown | QH1 | 0.777 | 0.850 | 0.587 |
| | QH2 | 0.800 | | |
| | QH3 | 0.764 | | |
| | QH4 | 0.722 | | |
| Perceived Benefits | PB1 | 0.723 | 0.885 | 0.562 |
| | PB2 | 0.731 | | |
| | PB3 | 0.810 | | |
| | PB4 | 0.739 | | |
| | PB5 | 0.750 | | |
| | PB6 | 0.740 | | |
| Perceived Trust | PT1 | 0.749 | 0.863 | 0.612 |
| | PT2 | 0.871 | | |
| | PT3 | 0.745 | | |
| | PT4 | 0.757 | | |
| Perceived Cost | PC1 | 0.805 | 0.878 | 0.589 |
| | PC2 | 0.774 | | |
| | PC3 | 0.721 | | |
| | PC4 | 0.764 | | |
| | PC5 | 0.774 | | |
| Support | ST1 | 0.780 | 0.8664 | 0.6186 |
| | ST2 | 0.786 | | |
| | ST3 | 0.787 | | |
| | ST4 | 0.793 | | |
| Government Support | GS1 | 0.815 | 0.8951 | 0.6307 |
| | GS2 | 0.787 | | |
| | GS3 | 0.774 | | |
| | GS4 | 0.812 | | |
| | GS5 | 0.782 | | |
| Intention to Return Home | IRH1 | 0.785 | 0.8631 | 0.6120 |
| | IRH2 | 0.800 | | |
| | IRH3 | 0.745 | | |
| | IRH4 | 0.798 | | |

attachment positively affects perceived trust (H6 supported, P<0.001, standardized path coefficient = 0.19). Place attachment has a significant positive effect on both perceived benefits and perceived cost (H7 and H8 supported, P<0.001, standardized path coefficients = 0.34 and 0.19, respectively). Community involvement significantly affects perceived benefits and perceived cost (H9 and H11 supported, P<0.001, standardized path coefficients = 0.36 and 0.34, respectively), but does not significantly affect perceived trust (H10 not supported, P=0.096). Quality of life in the hometown significantly affects perceived cost (H12 supported, P<0.001, standardized path coefficient = 0.26). Perceived benefits, trust, and cost all have significant

**Table 6. Discriminant validity.**

|  | FA | IRH | GS | ST | PC | PB | QLH | CI | PE | PT |
|---|---|---|---|---|---|---|---|---|---|---|
| **Family Attachment** | 0.835 |  |  |  |  |  |  |  |  |  |
| **Intention to Return Home** | -0.003 | 0.791 |  |  |  |  |  |  |  |  |
| **Government Support** | 0.268 | 0.232 | 0.806 |  |  |  |  |  |  |  |
| **Support** | 0.204 | 0.325 | 0.472 | 0.798 |  |  |  |  |  |  |
| **Perceived Cost** | 0.212 | 0.198 | 0.442 | 0.485 | 0.792 |  |  |  |  |  |
| **Perceived Benefits** | 0.189 | 0.155 | 0.455 | 0.491 | 0.509 | 0.803 |  |  |  |  |
| **Quality of Life in Hometown** | 0.268 | 0.069 | 0.409 | 0.310 | 0.420 | 0.464 | 0.834 |  |  |  |
| **Community Involvement** | 0.246 | 0.099 | 0.329 | 0.319 | 0.445 | 0.453 | 0.281 | 0.825 |  |  |
| **Place Emotion** | 0.060 | 0.137 | 0.443 | 0.339 | 0.341 | 0.435 | 0.289 | 0.216 | 0.793 |  |
| **Perceived Trust** | 0.237 | 0.211 | 0.459 | 0.573 | 0.451 | 0.529 | 0.465 | 0.109 | 0.357 | 0.831 |

**Table 7. Structural equation model path coefficient test.**

| Hyp. | Path | Esti-mate | S.E. | C.R. | P Label | Std. Coef. | Conclusion |
|---|---|---|---|---|---|---|---|
| H5 | Perceived Benefits ← Family Attachment | 0.073 | 0.046 | 1.572 | 0.116 | 0.07 | Not Supported |
| H6 | Perceived Trust ← Family Attachment | 0.189 | 0.056 | 3.347 | ∗∗∗ | 0.19 | Supported |
| H7 | Perceived Benefits ← Place Emotion | 0.338 | 0.057 | 5.916 | ∗∗∗ | 0.34 | Supported |
| H8 | Perceived Cost ← Place Emotion | 0.191 | 0.057 | 3.337 | ∗∗∗ | 0.19 | Supported |
| H9 | Perceived Benefits ← Community Involvement | 0.360 | 0.061 | 5.882 | ∗∗∗ | 0.36 | Supported |
| H10 | Perceived Trust ← Community Involvement | 0.108 | 0.065 | 1.664 | 0.096 | 0.11 | Not Supported |
| H11 | Perceived Cost ← Community Involvement | 0.337 | 0.062 | 5.483 | ∗∗∗ | 0.34 | Supported |
| H12 | Perceived Cost ← Quality of Life in Hometown | 0.263 | 0.066 | 3.985 | ∗∗∗ | 0.26 | Supported |
| H2 | Support ← Perceived Benefits | 0.213 | 0.061 | 3.507 | ∗∗∗ | 0.21 | Supported |
| H3 | Support ← Perceived Trust | 0.382 | 0.060 | 6.356 | ∗∗∗ | 0.38 | Supported |
| H4 | Support ← Perceived Cost | 0.248 | 0.060 | 4.123 | ∗∗∗ | 0.25 | Supported |
| H1 | Intention to Return Home ← Support | 0.345 | 0.074 | 4.677 | ∗∗∗ | 0.35 | Supported |

*Notes:* * p<0.05; ** p<0.01; *** p<0.001.

**Table 8. Results of government support moderating the relationship between perceived benefits and support.**

| Experimental Result | | | | |
|---|---|---|---|---|
| Model Path | Coefficient (Estimate) | Standard Error (S.E.) | Critical Ratio (C.R.) | P value |
| Constant | 3.6655 | 0.7316 | 5.0102 | 0.0000 |
| Perceived Benefits | -0.3163 | 0.2046 | -1.5463 | 0.1231 |
| Government Support | -0.3456 | 0.2057 | -1.6800 | 0.0940 |
| Perceived Benefits × Government Support | 0.1776 | 0.0544 | 3.2638 | 0.0012 |
| **Conditional Effects of Regulatory Effects** | | | | |
| Government Support | Effect | Standard Error (S.E.) | Critical Ratio (t) | P value | Lower Limit (LLCI) | Upper Limit (ULCI) |
| 2.7816 | 0.1777 | 0.0716 | 2.4828 | 0.0136 | 0.0369 | 0.3185 |
| 3.8058 | 0.3596 | 0.0560 | 6.4182 | 0.0000 | 0.2493 | 0.4698 |
| 4.8299 | 0.5415 | 0.0858 | 6.3085 | 0.0000 | 0.3726 | 0.7103 |

positive effects on support (H2, H3, and H4 supported, P<0.001, standardized path coefficients = 0.21, 0.38, and 0.25, respectively). Support has a significant positive effect on the intention to return home (H1 supported, P<0.001, standardized path coefficient = 0.35). In summary, most hypotheses are supported, indicating that factors like family attachment, place attachment, community involvement, and hometown quality of life influence perceived benefits, trust, and cost. These factors, in turn, affect support and ultimately influence migrant workers' intention to return home.

### 4.6 Moderation effect analysis

**4.6.1 Analysis of government support as a moderator in the relationship between perceived benefits and support.** Table 8 shows that the interaction between government support and perceived benefits has a significant moderating effect. The path coefficient is 0.1776, with a standard error of 0.0544, C.R of 3.2638, and a P-value of 0.0012, indicating significance below 0.01. This shows that perceived benefits' effect on support varies by the level of government support, significantly strengthening this link. The conditional effect analysis further confirms the moderating effect by examining perceived benefits' impact on support at varying levels of government support. At low government support (coefficient value of 2.7816), perceived benefits have a relatively weak positive effect on support (effect = 0.1777, SE = 0.0716, t-value = 2.4828, P = 0.0136). This P-value shows that the effect is significant but relatively small under low government support. The confidence interval (LLCI = 0.0369, ULCI = 0.3185) is positive but narrow, suggesting limited enhancement of support by perceived benefits. With increased government support (coefficient = 3.8058), the impact of perceived benefits on support strengthens significantly (effect = 0.3596, SE = 0.0560, t-value = 6.4182, P = 0.0000). The confidence interval (LLCI = 0.2493, ULCI = 0.4698) is positive and wider, indicating a stronger enhancing effect of perceived benefits on support. At high government support (coefficient = 4.8299), the effect of perceived benefits on support increases further (effect = 0.5415, SE = 0.0858, t-value = 6.3085, P = 0.0000). The confidence interval (LLCI = 0.3726, ULCI = 0.7103) expands, showing that high government support further strengthens the positive effect of perceived benefits on support.

**4.6.2 Analysis of government support as a moderator in the relationship between perceived trust and support.** Table 9 shows that the direct effect of perceived trust on support is significant. The path coefficient is 0.5681, with a standard error of 0.1981, C.R of 2.8672, and P-value of 0.0044. This shows that the positive effect of perceived trust on support is significant. Government support also has a significant positive impact on support. The path coefficient is 0.4334, with a standard error of 0.1817, critical ratio of 2.3855, and P-value of 0.0177. This indicates that government policies, funding, and services can directly enhance migrant workers' perception of support for returning home. However, the interaction between perceived trust and government support does not significantly impact support. The path coefficient is -0.0469, with a standard error of 0.0504, critical ratio of -0.9315, and P-value of 0.3523, indicating lack of significance. This indicates that government support does not significantly moderate the relationship between perceived trust and support.

**4.6.3 Analysis of government support as a moderator in the relationship between perceived cost and support.** Table 10 shows that the direct effect of perceived cost on support is not significant. The path coefficient is -0.2814, with a standard error of 0.1996, C.R of -1.4102, and P-value of 0.1595, indicating lack of significance. However, government support has a significant direct effect on support. The path coefficient is -0.3969, with a standard error

**Table 9. Results of government support moderating the relationship between perceived trust and support.**

| Experimental Result | | | | |
|---|---|---|---|---|
| **Model Path** | **Coefficient (Estimate)** | **Standard Error (S.E.)** | **Critical Ratio (C.R.)** | **P value** |
| Constant | 0.7014 | 0.6750 | 1.0392 | 0.2995 |
| Perceived Trust | 0.5681 | 0.1981 | 2.8672 | 0.0044 |
| Government Support | 0.4334 | 0.1817 | 2.3855 | 0.0177 |
| Perceived Trust × Government Support | -0.0469 | 0.0504 | -0.9315 | 0.3523 |

of 0.1942, critical ratio of -2.0437, and P-value of 0.0418, reaching significance. This suggests that government support can directly enhance migrant workers' perception of support. Additionally, the interaction between perceived cost and government support has a significant positive moderating effect on support. The path coefficient is 0.1410, with a standard error of 0.0521, critical ratio of 2.7067, and P-value of 0.0072. This indicates that government support significantly moderates the effect of perceived cost on support, enhancing migrant workers' perception of support by reducing perceived costs. In the conditional effect analysis, at low government support (coefficient = 2.7816), the effect of perceived cost on support is not significant. The effect value is 0.1108, with a standard error of 0.0739, t-value of 1.4989, and P-value of 0.1349, indicating lack of significance. The confidence interval (LLCI = -0.0347, ULCI = 0.2563) crosses zero, indicating that at low government support, the positive effect of perceived cost on support is minimal. At moderate government support (coefficient = 3.8058), the effect of perceived cost on support significantly strengthens. The effect is 0.2552, with a standard error of 0.0583, critical ratio of 4.3801, and P-value of 0.0000, reaching significance. The confidence interval (LLCI = 0.1406, ULCI = 0.3699) is fully positive. At high government support (coefficient = 4.8299), the effect of perceived cost on support continues to strengthen. The effect increases to 0.3996, with a standard error of 0.0838, critical ratio of 4.7707, and a highly significant P-value of 0.0000. The confidence interval (LLCI = 0.2348, ULCI = 0.5645) is fully positive and wider, indicating that high government support substantially alleviates concerns about return costs, enhancing migrant workers' perception of support for returning. The results show that government support has a significant positive moderating effect between perceived cost and support, particularly strong at medium to high levels of government support.

## 4.7 Discussion on variable correlation and multicollinearity

This study establishes a structural equation model of migrant workers' willingness to return home grounded in push-pull theory. It examines the influences of family attachment, local emotions, community participation, and hometown quality of life on migrant workers' perceived benefits, perceived trust, and perceived costs. Furthermore, it thoroughly discusses the correlation and multicollinearity issues among the various variables. Theoretically, hypothesis H5 proposes that family attachment can augment economic returns and emotional satisfaction in anticipation of returning home. However, empirical results reveal a weak direct economic effect, with a path coefficient for perceived benefits of 0.073 (P = 0.116), suggesting

**Table 10. Results of government support moderating the relationship between perceived cost and support.**

| Experimental result | | | | |
|---|---|---|---|---|
| Model Path | Coefficient (Estimate) | Standard Error (S.E.) | Critical Ratio (C.R.) | P value |
| Constant | 4.1828 | 0.6973 | 5.9987 | 0.0000 |
| Support | -0.2814 | 0.1996 | -1.4102 | 0.1595 |
| Government Support | -0.3969 | 0.1942 | -2.0437 | 0.0418 |
| Support × Government Support | 0.1410 | 0.0521 | 2.7067 | 0.0072 |
| **Conditional Effects of Regulatory Effects** | | | | |
| Government Support | Effect | Standard Error (S.E.) | Critical Ratio (t) | P value | Lower Limit (LLCI) | Upper Limit (ULCI) |
| 2.7816 | 0.1108 | 0.0739 | 1.4989 | 0.1349 | -0.0347 | 0.2563 |
| 3.8058 | 0.2552 | 0.0583 | 4.3801 | 0.0000 | 0.1406 | 0.3699 |
| 4.8299 | 0.3996 | 0.0838 | 4.7707 | 0.0000 | 0.2348 | 0.5645 |

its primary role lies in trust building. Both factor analysis and discriminant validity tests demonstrate high discrimination among constructs such as family attachment, local emotions, and community participation, with no severe multicollinearity observed. The H6 results show a significant positive impact of family attachment on perceived trust (standardized path coefficient = 0.19, P < 0.001). Similarly, H7 indicates that local emotions significantly enhance perceived benefits (standardized path coefficient = 0.34, P < 0.001), and both are strictly distinguished. H8 finds that local emotions also positively affect perceived costs (standardized path coefficient = 0.19, P < 0.001), highlighting migrant workers' heightened sensitivity to the costs of returning home when deeply emotionally involved. Regarding community participation, H9 demonstrates a significant positive influence on perceived benefits (standardized path coefficient = 0.36, P < 0.001), whereas its impact on perceived trust is insignificant in H10 (path coefficient = 0.108, P = 0.096), suggesting that mere participation is insufficient for trust enhancement. H11 shows that community participation significantly increases perceived costs (standardized path coefficient = 0.34, P < 0.001). Additionally, H12 indicates that hometown quality of life has a significant positive effect on perceived costs (standardized path coefficient = 0.26, P < 0.001).

In terms of the impact of perceived factors on support, H2, H3, and H4 respectively demonstrate significant positive effects of perceived benefits, perceived trust, and perceived costs on support (standardized path coefficients of 0.21, 0.38, and 0.25, respectively, P < 0.001). Support, as a mediating variable, also significantly influences the willingness to return home (standardized path coefficient = 0.35, P < 0.001). Factor analyses, internal consistency, and discriminant validity tests confirm the good independence of the constructs. Regarding the moderating effect of government support, H3a shows a significant interaction between government support and perceived benefits (path coefficient = 0.1776, P = 0.0012), while H3b reveals no significant interaction between government support and perceived trust (path coefficient = -0.0469, P = 0.3523). Furthermore, H3c indicates that government support significantly mitigates the impact of perceived costs on support (interaction term path coefficient = 0.1410, P = 0.0072). Overall, rigorous questionnaire design, factor extraction, and discriminant validity testing ensure the independence of each construct and the stability of model estimation. Despite potential theoretical relationships among variables, empirical testing reveals no serious multicollinearity issues. This provides a foundation for future quantitative testing methods, such as variance inflation factor (VIF) and tolerance and suggests directions for improving related research.

## 4.8 Discussion

This study conducted a comprehensive path analysis of factors influencing migrant workers' intentions to return home, with a detailed examination of the moderating effect of government support. The hypotheses were tested using exploratory factor analysis, confirmatory factor analysis, and structural equation modeling. As shown in Fig 3, most hypotheses were supported, consistent with the "Push-Pull Theory" in migration studies. Perceived benefits, trust, and cost were all confirmed as significant factors. Migrant workers' perception of economic benefits in their hometown strengthens their perception of return support, aligning with pull factors that increase the attractiveness of returning. The effect of perceived trust on support suggests that trust in family, community, and government enhances migrant workers' willingness to return. The effect of perceived cost on support suggests that reducing return costs increases migrant workers' recognition of support. Reducing push factors is a key driver in the decision to return. Additionally, the study found that family attachment, place attachment, community involvement, and hometown quality of life influence perceived benefits, trust,

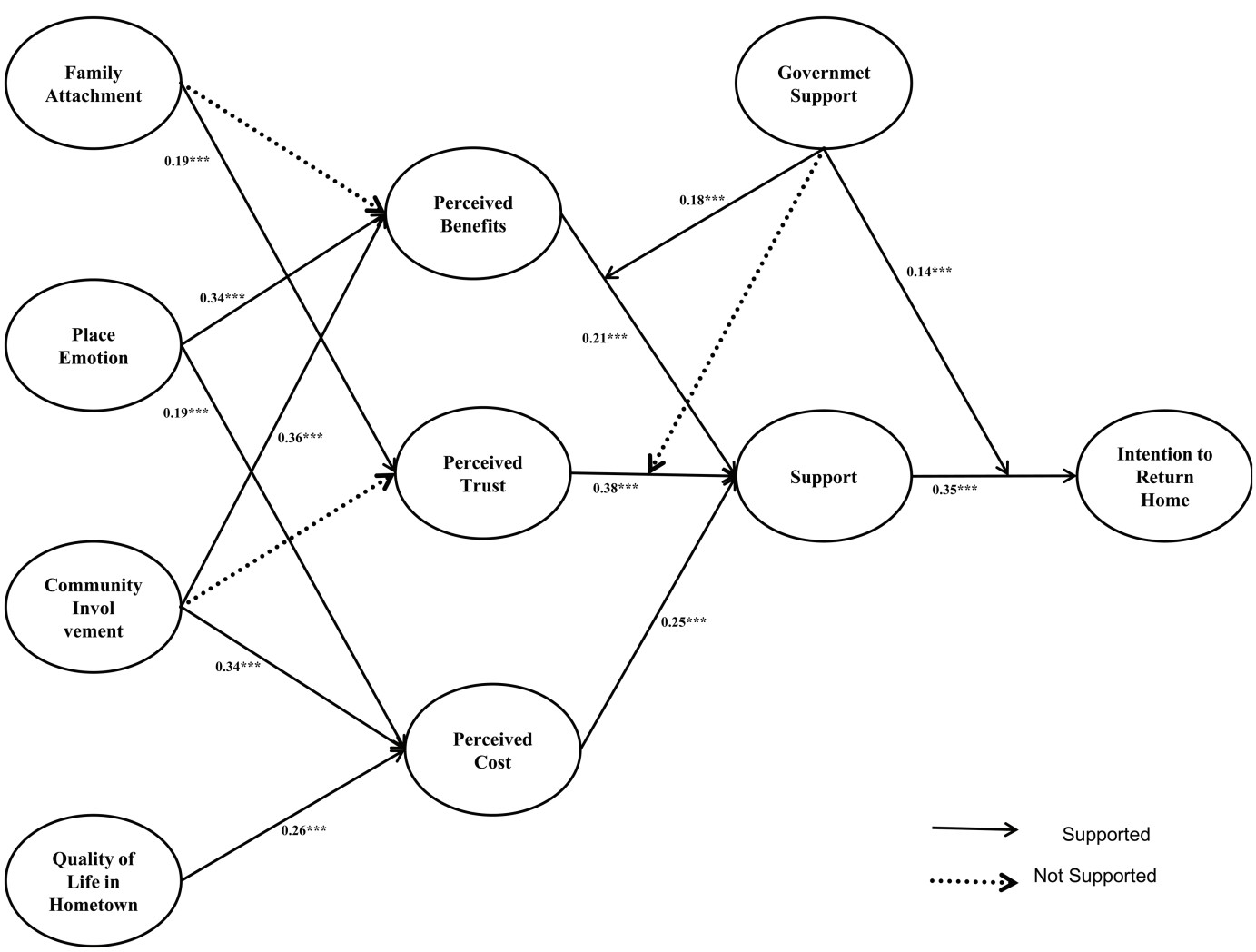

**Fig 3. Path coefficients of hypotheses.**

and cost. By enhancing emotional connections to their hometown and perceptions of support, these factors further motivate the decision to return. The empirical results indicate that government support significantly moderates the effects of perceived benefits and costs on return support. Specifically, high levels of government support enhance the positive impact of perceived benefits and reduce the negative impact of perceived costs, thereby strengthening return support. This finding underscores the importance of policy interventions, such as subsidies, training programs, and local job creation initiatives, in amplifying the economic appeal of returning home. However, government support does not significantly moderate the relationship between perceived trust and return support. This suggests that policy interventions alone may not substantially influence trust levels among migrant workers. Instead, trust appears to be more deeply rooted in social capital, historical experiences, and personal networks, which are less directly affected by government actions. This interpretation aligns with findings from Zhu et al. (2025), who emphasize the role of social networks in shaping migrant workers' trust and adaptation processes [55]. This contrasts with findings from other migration contexts, where policy interventions have been shown to directly influence trust

levels. For example, in some international migration studies, government programs aimed at fostering integration have successfully built trust among migrant populations. The divergence in our results underscores the need for context-specific approaches to migration policy [56].

In the path analysis, the coefficient for perceived trust to support is relatively high (0.38), indicating trust as a key factor influencing support. This finding aligns with previous research highlighting social capital's importance in individual decision-making [57]. The path coefficient for perceived benefits to support is 0.21. Though slightly lower than that for trust, it suggests that economic benefits are crucial in influencing return decisions. This result aligns with Bucheli et al. (2022), who emphasized the central role of economic motivation in migration decisions [58]. The path coefficient for perceived cost to support is 0.25, indicating that reducing costs enhances perceived support. This supports Nguyen's (2023) view on push factors in the Push-Pull Theory, suggesting that reducing resistance is a key driver in migration decisions [59]. Additionally, government support significantly moderates the relationships between perceived benefits and support and perceived cost and support, but not between perceived trust and support. This suggests that government policies addressing economic and cost factors can increase return willingness, but additional efforts are needed to foster trust. From a theoretical perspective, the study advances the Push-Pull Theory by integrating emotional ties and government support as moderating factors, providing a more nuanced understanding of return migration dynamics. The results highlight the interplay between economic incentives, emotional bonds, and institutional trust, offering a holistic framework for analyzing migration decisions. This refined framework is particularly relevant in the context of China's rapid urbanization and rural revitalization efforts, where dynamic push-pull factors shape migration patterns. From a policy perspective, the findings suggest that governments should prioritize economic incentives and cost-reduction measures to encourage return migration. For instance, policies that enhance local economic opportunities, provide subsidies for rural entrepreneurship, and reduce barriers to return (e.g., housing and transportation costs) can significantly increase return willingness. However, the limited moderating effect of government support on trust underscores the need for complementary efforts to build trust in institutions. Enhancing transparency, improving policy implementation, and fostering community engagement are critical steps in this direction.

This study found two noteworthy non significant relationships: firstly, there was no significant correlation between family emotional connections (H5) and expected returns to hometown. This may be due to two reasons: on the one hand, the "expected returns" in the research mainly measure economic returns, while family emotions belong to non economic factors, and the two dimensions are different; On the other hand, the survey found that many migrant workers choose to return to their hometowns even though they know their income is low due to family responsibilities, indicating that family factors influence decision-making more through emotional obligations rather than rational calculations. Secondly, there was no expected correlation between community participation (H10) and community trust. This reflects the particularity of contemporary rural China: although community participation traditionally enhances trust, the long-term absence of migrant workers makes this mechanism ineffective. Data shows that many returnees still feel difficult to integrate even when participating in village affairs activities, indicating that short-term formal participation is difficult to rebuild deep trust relationships [60]. These findings emphasize the necessity of rethinking how traditional concepts of social capital can be applied to the rapidly changing context of rural to urban migration in China. Future research should explore potential interactive effects, such as whether family attachment only becomes important when combined with certain economic conditions, or whether specific types of community participation are more effective in rebuilding trust among returnees.

From a broader perspective, this study offers insights for businesses and practitioners on innovating to support rural revitalization. The lack of a confirmed moderating effect of perceived trust on support suggests limited effectiveness in relying solely on government policies to build trust. This opens opportunities for businesses and practitioners to innovate and make an impact. First, companies can address the trust gap through corporate social responsibility (CSR) initiatives. For example, they can support returning entrepreneurs and provide skills training to increase migrant workers' trust in companies and the community [61]. Second, businesses should focus on meeting the needs of returning workers through innovative products and services. This includes developing flexible work formats, improving infrastructure, and creating family-friendly jobs to enhance employment stability. Additionally, practitioners and local businesses should focus on community collaboration. They should promote community development, strengthen workers' sense of belonging, and foster emotional ties to their hometowns through involvement [62]. Thus, businesses and practitioners should innovate in technology, social responsibility, and community collaboration to bridge trust gaps left by government policies. This approach can lead to a win-win outcome for economic and social benefits [63]. By doing so, businesses can support rural revitalization while achieving sustainable development and building a solid social capital network. This provides broader support for returning workers, better serving rural revitalization goals.

## 5 Implications

### 5.1 Theoretical implications

This study constructs a comprehensive model of migrant workers' return intentions based on the Push-Pull Theory. The model includes factors like perceived benefits, trust, cost, and support, and verifies the moderating role of government support. This analysis of return intentions leads to the following theoretical insights. First, this study confirms that the Push-Pull Theory is an effective framework for understanding migration and return behaviors among migrant workers. The application of the Push-Pull Theory offers support in understanding the driving and restraining forces in migration decisions, particularly concerning economic benefits, emotional bonds, and costs [64]. Second, this study broadens the application of the Push-Pull Theory in return intention research. It incorporates emotional factors like family attachment, place attachment, and community involvement into the model. This highlights that, beyond economic factors, emotional bonds and social connections also play key roles in migration decisions. This aligns with Cruz-Manjarrez (2023) on the importance of social networks in economic behavior. Thus, integrating emotional bonds and social capital perspectives can better explain the complexity of migrant workers' behaviors [65]. Additionally, this study finds that government support significantly moderates perceived benefits and costs. This aligns with Hormozinejad (2023) on the importance of corporate responsibility and government policies in socioeconomic development [66]. It suggests that policy incentives reduce economic costs and increase return intentions by enhancing perceptions of hometown benefits. This study extends the Push-Pull Theory by incorporating diverse factors and moderating variables, enhancing the theory's applicability and explanatory power in the context of migration in China. It provides a comprehensive foundation for future research in this field.

### 5.2 Practical implications

This study offers practical insights for governments, businesses, and communities to formulate policies that encourage migrant workers to return home. First, the findings indicate

that government support significantly reduces perceived costs and increases perceived benefits. This suggests that local governments should enhance policy incentives, including return-to-home employment subsidies and funding for entrepreneurship. These measures can reduce migrant workers' economic burden and boost their confidence in hometown employment and living conditions [67]. Second, community involvement and place attachment are also important factors influencing return intentions. Therefore, local governments and communities should focus on community building to strengthen cohesion and belonging. Community activities and improved public services can deepen migrant workers' emotional connections to their hometowns [68]. Additionally, businesses play a crucial role in rural revitalization. They should encourage migrant workers' return by offering flexible job opportunities and professional training. This supports returning workers in adjusting to work and living conditions in their hometowns [69]. A multi-level, multi-dimensional support system can promote migrant workers' return, supporting rural revitalization and urban-rural coordination. This study provides a practical foundation for governments, businesses, and communities in policy formulation and implementation. It highlights the role of economic incentives, community building, and business involvement in encouraging migrant workers to return.

## 6 Limitations and future research

This study has several limitations that provide opportunities for future research. First, this study only focuses on migrant workers in Hunan Province, which may limit the generalizability of the research results to other regions in China. Given China's vast geographical and socio-economic diversity, factors influencing the decision to return home for immigration may vary by province. For example, coastal provinces with different economic structures and labor market conditions may exhibit unique migration patterns that were not captured in this study. Although Hunan has become an important case study due to its position as a major labor export region, the specific economic conditions, cultural norms, and policy environment of other provinces may produce different results. Future research will benefit from comparative studies across multiple provinces to better understand how regional characteristics regulate the relationship between economic factors, emotional connections, and return migration decisions. Second, the use of cross-sectional data restricts our ability to capture dynamic changes in return intentions over time. While structural equation modeling (SEM) provides robust analytical insights, it remains a correlational method. To address this, future research could adopt a longitudinal design to examine the long-term trends and trajectories of return migration. Additionally, robustness checks, such as instrumental variable approaches or quasi-experimental designs, could help mitigate potential endogeneity issues and provide stronger causal evidence. Third, this study does not explicitly incorporate land ownership as a factor influencing return migration decisions. Land ownership may serve as a critical economic asset that either encourages return migration by providing security or sustains migration elsewhere by offering financial stability. Future research should include land ownership as a control variable or moderator to explore its role in shaping return intentions. Furthermore, the impact of the current economic downturn on return migration intentions warrants investigation, as economic instability may alter the push-pull dynamics for migrant workers. Finally, the discussion of findings could be expanded to compare the results more explicitly with existing literature. For instance, how do the observed relationships between economic incentives, emotional ties, and government support align with or diverge from previous studies on return migration in other contexts? Such comparisons could provide deeper insights into the unique characteristics of Chinese return migration and inform context-specific policy

recommendations. The findings of this study have important practical implications for policymakers and businesses. For policymakers, the results highlight the need for targeted interventions that address both economic incentives and emotional factors. For example, local governments could develop programs that enhance economic opportunities in rural areas while also fostering community engagement and family support. For businesses, understanding the factors influencing return migration can inform strategies for workforce retention and regional development.

## 7 Conclusion

This study constructed a comprehensive model based on the Push-Pull Theory to explore key factors influencing migrant workers' intentions to return home and their interrelationships. Perceived benefits, trust, and cost positively affect support, which is crucial in encouraging migrant workers to return. This suggests that migrant workers consider both economic benefits and emotional bonds when deciding to return. Government support significantly moderates the relationships between perceived benefits, cost, and support, indicating that policy incentives, economic assistance, and cost reduction measures enhance support. However, government support does not significantly moderate the relationship between trust and support. This implies that policy incentives alone may not increase trust; improving transparency and public service quality is essential. Additionally, family attachment, place attachment, and community involvement indirectly influence return decisions by affecting perceived benefits, trust, and cost. This highlights the importance of emotional factors and social connections in migration behavior. This study extends the Push-Pull Theory by providing a comprehensive perspective on migrant workers' return behaviors. It also offers policy recommendations, emphasizing economic incentives, community building, and trust enhancement for governments, businesses, and communities. Future research could expand sample coverage and employ longitudinal studies to explore the complex roles of additional external factors. This could support coordinated urban-rural development and rural revitalization.

## Supporting information

**S1 Text. Questionnaire.**
(PDF)

**S2 Text. RawData.**
(XLSX)

## Author contributions

**Conceptualization:** Jun Xu.

**Data curation:** Qiong Zhou, Zhe Huang.

**Formal analysis:** Qiong Zhou, Lining Zeng, Zhe Huang, Jun Xu.

**Funding acquisition:** Qiong Zhou, Lining Zeng.

**Investigation:** Zhe Huang.

**Methodology:** Qiong Zhou.

**Project administration:** Lining Zeng, Jun Xu.

**Resources:** Zhe Huang.

**Supervision:** Zhe Huang, Jun Xu.

**Validation:** Lining Zeng.

**Visualization:** Jun Xu.

**Writing – original draft:** Lining Zeng.

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
