## [Decision Letter · Decision Letter 0]

20 Jun 2025

PONE-D-24-52390Why Do Chinese Migrant Workers Return? Exploring Economic Push-Pull Factors and Emotional TiesPLOS ONE

Dear Dr. Zeng,

Thank you for submitting your manuscript to PLOS ONE. After careful consideration, we feel that it has merit but does not fully meet PLOS ONE’s publication criteria as it currently stands. Therefore, we invite you to submit a revised version of the manuscript that addresses the points raised during the review process.

Please reflect deeply on the reviewer's comments to enhance the rigor of the paper, especially the representativeness of the sample.Please conduct a thorough analysis of the references recommended by the reviewers. Only those that are valuable to this study can be cited, not all of them must be cited.

We look forward to receiving your revised manuscript.

Kind regards,

Bifeng Zhu

Academic Editor

PLOS ONE

4. Please ensure that you refer to Figure 1 and 2 in your text as, if accepted, production will need this reference to link the reader to the figure.

Additional Editor Comments (if provided):

Reviewers' comments:

Reviewer's Responses to Questions

**Comments to the Author**

1. Is the manuscript technically sound, and do the data support the conclusions?

Reviewer #1: Partly

Reviewer #2: Yes

2. Has the statistical analysis been performed appropriately and rigorously? 

Reviewer #1: Yes

Reviewer #2: Yes

3. Have the authors made all data underlying the findings in their manuscript fully available?

Reviewer #1: Yes

Reviewer #2: Yes

4. Is the manuscript presented in an intelligible fashion and written in standard English?

Reviewer #1: Yes

Reviewer #2: Yes

5. Review Comments to the Author

Reviewer #1: This study examines the factors influencing the return migration of Chinese migrant workers, an increasingly relevant issue given recent economic and policy developments. The authors employ the Push-Pull Theory to develop a comprehensive model, incorporating perceived benefits, trust, costs, and support (both personal and governmental). Using structural equation modeling (SEM), the study finds that perceived benefits, trust, and costs significantly shape migrant workers' perception of return support, subsequently affecting their willingness to return. Additionally, government support moderates the relationships between perceived benefits and costs with return support but does not significantly affect the link between perceived trust and support. The study further highlights the role of emotional factors—such as family attachment, community involvement, and quality of life—in shaping economic perceptions and return intentions.

The paper addresses a timely and essential issue in migration studies and contributes to the literature by integrating economic incentives with emotional factors in return migration decisions. The application of SEM provides a rigorous empirical approach to testing the proposed model. However, some areas require further clarification and improvement to enhance the study’s theoretical contributions, methodological rigor, and practical implications.

Major comments

The authors should discuss the motivation of this study more explicitly. The authors should clarify how this study differs from previous studies and explicitly articulate its unique contributions in the introduction. For instance, how this paper differs from Bi et al. 92019) discussed the migrant workers' social networking and Fan et al. (2025) on land ownership and migrants’ settlement and maximizing the labor economy. So, how is this study different from previous studies? Clearly outlining these aspects will enhance the paper’s impact and contextual significance.

References:

https://doi.org/10.1016/j.habitatint.2019.02.008

https://doi.org/10.1016/j.landusepol.2024.107411

While the Push-Pull Theory is well established, a more explicit discussion of how this study extends or refines the theory would strengthen its contribution. The manuscript should clearly articulate how integrating emotional bonds enhances the explanatory power of the Push-Pull framework.

Some key concepts, such as "perceived trust" and "return support," need clearer definitions and justification in terms of measurement. Providing a detailed explanation of how these variables were operationalized would improve clarity.

The paper should discuss how representative the sample is of the broader population of migrant workers in China. Were specific regions, industries, or demographic groups over- or underrepresented? This is not shown in Figure 2. Addressing this would improve the generalizability of the findings.

Lines 330-331 “Data collection occurred from March to July 2024 across various cities in Hunan Province, China, resulting in 312 valid responses…”

This dataset focuses on migrant workers from Hunan. How can it be generalised to Chinese migrant workers? A qualification is required if it can be generalised.

Why Hunan? Are there any dedicated reasons to select migrant workers from Hunan?

Figure 2 is interesting. Why do migrant workers prefer cities like Guangzhou and Shenzhen over Beijing and Shanghai? Is it common for Hunan workers not to go to the largest two Chinese cities?

While SEM provides robust analytical insights, it remains a correlational method. The paper should be cautious about causal claims and consider potential endogeneity issues. If possible, robustness checks (e.g., instrumental variable approaches or longitudinal data) should be discussed as avenues for future research.

The empirical results indicate that government support significantly moderates the relationship between perceived benefits and support but does not moderate the relationship between perceived trust and support. The authors should further discuss the theoretical and practical implications of this finding. Why might government support amplify the role of perceived benefits but not influence trust? Does this suggest trust is developed through other means, such as social networks or past experiences, rather than policy interventions? See Bi et al. (2019) in Habitat International.

The finding that government support does not moderate the relationship between trust and support is essential but lacks a deeper interpretation. The authors should consider potential explanations—perhaps trust is more influenced by social capital or historical experiences rather than government incentives. Additionally, the authors could compare this result with findings from other migration studies to understand whether this is a unique characteristic of Chinese return migration.

Additionally, land ownership could be an essential factor influencing return migration decisions. See Fan et al. (2025) in Land Use Policy. A robustness check incorporating land ownership as a control variable or moderator would provide further insights into whether economic security through land assets encourages return migration or sustains migration elsewhere. Further, would the current economic downturn impact the intention of their return? It would be interesting to check this point.

The discussion section could be expanded to compare the findings with existing literature more explicitly. How do these results align with or diverge from previous studies on return migration? Additionally, practical implications for policymakers and businesses could be further elaborated.

Overall, this study contributes to understanding return migration in China. The integration of economic and emotional factors into the Push-Pull framework is a key strength. However, improvements in the motivation, theoretical clarity, variable definitions, sample representativeness, and discussion of causality would enhance the study’s rigor and impact. Addressing these points will further solidify the paper’s contribution to migration studies and policy discussions.

Reviewer #2: This study addresses an interesting and topical issue, namely worker migration. Applying a complex structural equation modelling to examine the interactions between perceived benefits, trust and perceived costs and how this influence migrant workers’ intention to return home, the paper emphasizes the moderating role of government support. Methodologically, the study employs a rigorous combination of exploratory and confirmatory factor analysis, along with structural equation modeling, to test the validity and reliability of theoretical constructs and examine the relationships among key variables in a representative sample of migrant workers in China.

My specific observations and recommendations for this study are:

1. Section Introduction: a) Mentions of specific reports and statistics presented need to be integrated in the context of the discussion. How are the demographic changes of migrant workers correlated with economic changes in different regions of China? b) The inclusion of some details about the Push-Pull theory methodology: how is this theory adapted to study the return of migrant workers, especially since it is mentioned that its applications in this context are limited; c) The last part of the introduction refers to government involvement as a moderating variable, but what is the specificity of the policies analyzed and how can these policies influence migrant workers' decisions to return?

2. In the Theoretical Background Section: a) introducing concrete examples of previous studies using Push-Pull Theory to contextualize the theoretical discussions in the literature; b) addressing the role of technology and social networks. Exploring how these aspects influence the migration of migrant workers could provide a more up-to-date perspective.

3. The "Hypotheses" section presents the conceptual relationships between different variables in the context of the proposed research model. Suggested additions and clarifications: a) Make sure that all key terms such as "perceived cost", "perceived trust", and "perceived benefits" are clearly defined; c) Provide details on how the variables involved in the hypotheses will be measured: the types of scales used, the number of items on each scale, or how the relationships are quantified; d) Conclude the section with a discussion of the potential implications of confirming the proposed hypotheses. How might the results influence migration theory or public policy?

4. This presented section, "Experimental Results" provides extensive details about the research design, data analysis, and interpretation of the results, which are essential for validating the study hypotheses. The following comments may help to improve and clarify the content: a) Details about the Data Collection Process: limitations of the data collection method, such as response bias or response rate; b) The detailed description of the factor analysis and the model fit analysis is nicely done. I recommend including some discussion of how the different variables are correlated and whether there are multicollinearity issues.

6. PLOS authors have the option to publish the peer review history of their article (what does this mean?). If published, this will include your full peer review and any attached files.

Reviewer #1: **Yes: **Chyi Lin Lee

Reviewer #2: No

---

## [Author Response · Author response to Decision Letter 1]

8 Mar 2025

Dear Reviewer,

Thank you for your thoughtful and constructive feedback on our manuscript titled “Why Do Chinese Migrant Workers Return? Exploring Economic Push-Pull Factors and Emotional Ties”. This study utilizes the Push-Pull Theory from migration theory to develop a comprehensive model. This model investigates how perceived benefits, trust, costs, and both personal and government support affect migrant workers’ willingness to return. It provides novel insights into how both economic incentives and emotional ties drive migrant workers’ decisions to return, offering a more nuanced understanding of migration dynamics in China. We greatly appreciate your time and effort in reviewing our work and providing detailed comments to help us improve the paper. Below, we address each of your major comments and outline the revisions we have made in response to your suggestions. In addition, we also uploaded all the raw data from this study to a publicly available database for viewing, Mendeley Data (Reserved DOI: 10.17632/s7ttjys754.1).

We believe that the revisions made in response to your comments have significantly strengthened the manuscript. The integration of specific data, expanded theoretical discussions, and detailed methodological explanations provide a more robust foundation for our study. We are confident that these improvements will enhance the clarity, relevance, and impact of our work. Thank you once again for your valuable feedback. We look forward to your response and hope that the revised manuscript meets the high standards of PLOS ONE.

Sincerely,

Zhou Qiong

School of Public Administration, Hunan Agricultural University, Changsha, Hunan, China

---

## [Decision Letter · Decision Letter 1]

1 Apr 2025

PONE-D-24-52390R1Why Do Chinese Migrant Workers Return? Exploring Economic Push-Pull Factors and Emotional TiesPLOS ONE

Dear Dr. Zeng,

Thank you for submitting your manuscript to PLOS ONE. After careful consideration, we feel that it has merit but does not fully meet PLOS ONE’s publication criteria as it currently stands. Therefore, we invite you to submit a revised version of the manuscript that addresses the points raised during the review process.

Provide necessary evidence or explanations based on the reviewer's comments.Please evaluate the reference recommended by reviewers reasonably. Note that it is not necessary to cite unless it actually contributes.

We look forward to receiving your revised manuscript.

Kind regards,

Bifeng Zhu

Academic Editor

PLOS ONE

Journal Requirements:

Reviewers' comments:

Reviewer's Responses to Questions

**Comments to the Author**

1. If the authors have adequately addressed your comments raised in a previous round of review and you feel that this manuscript is now acceptable for publication, you may indicate that here to bypass the “Comments to the Author” section, enter your conflict of interest statement in the “Confidential to Editor” section, and submit your "Accept" recommendation.

Reviewer #1: All comments have been addressed

2. Is the manuscript technically sound, and do the data support the conclusions?

Reviewer #1: Yes

3. Has the statistical analysis been performed appropriately and rigorously? 

Reviewer #1: Yes

4. Have the authors made all data underlying the findings in their manuscript fully available?

Reviewer #1: Yes

5. Is the manuscript presented in an intelligible fashion and written in standard English?

Reviewer #1: Yes

6. Review Comments to the Author

Reviewer #1: The paper satisfactorily addresses my previous comments. It is well-structured and contributes meaningfully to the literature. Integrating emotional bonds—such as family attachment, community involvement, and a sense of belonging—into the Push-Pull Theory is a valuable extension that enhances the understanding of migration decisions.

Minor Comments

1) The manuscript claims that living costs in Beijing and Shanghai are higher than in Guangzhou and Shenzhen. To strengthen this assertion, the authors should provide supporting statistical evidence, such as cost-of-living indices, rent data, or other economic indicators.

2) The study’s extension of the Push-Pull Theory to include emotional bonds is well-argued. A recent study by Lee et al. (2025) on Chinese migrant settlements in Greater Sydney offers empirical support for the role of social networks in shaping location preferences. Specifically, Chinese migrants prefer areas with larger Chinese communities, as these provide social engagement opportunities and facilitate settlement. The authors may consider referencing the following study to support this point in the theoretical background.

https://doi.org/10.1016/j.habitatint.2025.103331

The paper is in good shape overall, and only minor revisions are needed. The inclusion of statistical data for cost-of-living comparisons and a reference to the recent study on Chinese migrant settlement will further strengthen the manuscript. I look forward to seeing these refinements in the revised version.

7. PLOS authors have the option to publish the peer review history of their article (what does this mean?). If published, this will include your full peer review and any attached files.

Reviewer #1: No

---

## [Author Response · Author response to Decision Letter 2]

15 Apr 2025

Dear Reviewers,

Thank you for your constructive feedback on our manuscript. We sincerely appreciate the time and effort you have dedicated to reviewing our work. We have carefully considered all the comments and have revised the manuscript accordingly. Below, we provide a point-by-point response to the reviewers’ suggestions.

We believe these revisions have enhanced the rigor and clarity of our manuscript. Thank you again for your valuable input. Please find our revised manuscript attached, with all changes highlighted for ease of review.

Sincerely,

Zhou Qiong

Hunan Agricultural University, Hunan, China

---

## [Decision Letter · Decision Letter 2]

28 Apr 2025

PONE-D-24-52390R2Why Do Chinese Migrant Workers Return? Exploring Economic Push-Pull Factors and Emotional TiesPLOS ONE

Dear Dr. Zeng,

Thank you for submitting your manuscript to PLOS ONE. After careful consideration, we feel that it has merit but does not fully meet PLOS ONE’s publication criteria as it currently stands. Therefore, we invite you to submit a revised version of the manuscript that addresses the points raised during the review process.

The comments raised by the reviewer need to be explained in detail in the manuscript to enhance the rigor of the study.

We look forward to receiving your revised manuscript.

Kind regards,

Bifeng Zhu

Academic Editor

PLOS ONE

Journal Requirements:

Reviewers' comments:

Reviewer's Responses to Questions

**Comments to the Author**

1. If the authors have adequately addressed your comments raised in a previous round of review and you feel that this manuscript is now acceptable for publication, you may indicate that here to bypass the “Comments to the Author” section, enter your conflict of interest statement in the “Confidential to Editor” section, and submit your "Accept" recommendation.

Reviewer #1: All comments have been addressed

Reviewer #3: (No Response)

2. Is the manuscript technically sound, and do the data support the conclusions?

Reviewer #1: Yes

Reviewer #3: Partly

3. Has the statistical analysis been performed appropriately and rigorously? 

Reviewer #1: Yes

Reviewer #3: Yes

4. Have the authors made all data underlying the findings in their manuscript fully available?

Reviewer #1: Yes

Reviewer #3: Yes

5. Is the manuscript presented in an intelligible fashion and written in standard English?

Reviewer #1: Yes

Reviewer #3: Yes

6. Review Comments to the Author

Reviewer #1: Thank you — the overall content looks fine. However, the clean version that was uploaded is not the updated version reflected in the tracked changes. I’ve reviewed the tracked changes and am happy with the revisions, but please ensure that the final, clean version matches these updates.

Conditional acceptance is granted pending the upload of the correct final version.

Reviewer #3: This paper explores the factors driving Chinese migrant workers’ return migration decisions, integrating economic push-pull factors with emotional attachments. It provides valuable empirical insights and offers important policy implications for rural revitalization strategies. However, after carefully reviewing the manuscript, I believe that several clarifications and substantial revisions are needed to improve the overall quality of the paper.

1.The construct of "support" is central to the analysis but remains ambiguously defined at some points. Please clearly define "support" early in the conceptual framework and maintain a consistent interpretation throughout the paper.

2.The representativeness of the sample is questionable. The manuscript does not clearly explain how respondents were selected or whether a random sampling method was used. Please clarify the sample selection procedure and discuss potential biases arising from non-random sampling.

3.Return migration decisions are influenced by both destination and origin factors. Since the study’s sample is limited to Hunan Province, regional variations are not considered, which may affect the generalizability of the findings. Please explicitly acknowledge this limitation and discuss how regional differences could impact return migration behavior.

4.Some hypotheses (e.g., H5 and H10) were not supported, but no interpretation or explanation is provided. Please briefly discuss plausible reasons for these nonsignificant results in the discussion section.

5.The manuscript lists too many keywords. Please limit the keywords to 4–6 to align with common academic publishing standards and enhance focus.

6.The manuscript contains too many subdivisions, resulting in fragmented sections and occasional disruptions to the logical flow. Please consider simplifying the chapter structure by merging related sections and reducing the number of headings to improve coherence and readability.

7. PLOS authors have the option to publish the peer review history of their article (what does this mean?). If published, this will include your full peer review and any attached files.

Reviewer #1: No

Reviewer #3: No

---

## [Author Response · Author response to Decision Letter 3]

15 May 2025

Dear Reviewers,

We sincerely appreciate your time and constructive feedback on our manuscript. Your comments have significantly helped us improve the clarity, rigor, and coherence of the paper. This study aims to systematically investigate the factors driving the increasing trend of return migration among Chinese migrant workers by integrating economic and emotional perspectives within the framework of Push-Pull Theory. The study seeks to advance migration theory by developing a comprehensive model that bridges the gap between economic and emotional determinants of return migration, while also providing practical insights for rural revitalization policies by demonstrating that effective strategies must address both material incentives and trust-building measures. Below, we address each of your suggestions in detail and outline the revisions made in response. We believe these revisions have strengthened the manuscript significantly. Thank you again for your valuable input. Please let us know if further adjustments are needed.

Sincerely,

Zhou Qiong

Hunan Agricultural University, Hunan, China

---

## [Decision Letter · Decision Letter 3]

20 May 2025

Why Do Chinese Migrant Workers Return? Exploring Economic Push-Pull Factors and Emotional Ties

PONE-D-24-52390R3

Dear Dr. Zeng,

We’re pleased to inform you that your manuscript has been judged scientifically suitable for publication and will be formally accepted for publication once it meets all outstanding technical requirements.

Kind regards,

Bifeng Zhu

Academic Editor

PLOS ONE

Additional Editor Comments (optional):

Reviewers' comments:

Reviewer's Responses to Questions

**Comments to the Author**

1. If the authors have adequately addressed your comments raised in a previous round of review and you feel that this manuscript is now acceptable for publication, you may indicate that here to bypass the “Comments to the Author” section, enter your conflict of interest statement in the “Confidential to Editor” section, and submit your "Accept" recommendation.

Reviewer #1: All comments have been addressed

Reviewer #3: All comments have been addressed

2. Is the manuscript technically sound, and do the data support the conclusions?

Reviewer #1: Yes

Reviewer #3: (No Response)

3. Has the statistical analysis been performed appropriately and rigorously? 

Reviewer #1: Yes

Reviewer #3: (No Response)

4. Have the authors made all data underlying the findings in their manuscript fully available?

Reviewer #1: Yes

Reviewer #3: (No Response)

5. Is the manuscript presented in an intelligible fashion and written in standard English?

Reviewer #1: Yes

Reviewer #3: (No Response)

6. Review Comments to the Author

Reviewer #1: I am satisfied with the revision. My previous comments have been adequately addressed. I recommend acceptance.

Reviewer #3: (No Response)

7. PLOS authors have the option to publish the peer review history of their article (what does this mean?). If published, this will include your full peer review and any attached files.

Reviewer #1: No

Reviewer #3: No

---

## [Editor Report · Acceptance letter]

PONE-D-24-52390R3

PLOS ONE

Dear Dr. Zeng,

I'm pleased to inform you that your manuscript has been deemed suitable for publication in PLOS ONE. Congratulations! Your manuscript is now being handed over to our production team.

Kind regards,

on behalf of

Dr. Bifeng Zhu

Academic Editor

PLOS ONE